
# Comparative measurements of ambient atmospheric concentrations of ice nucleating particles using multiple immersion freezing methods and a continuous flow diffusion chamber

Paul J. DeMott[1], Thomas C. J. Hill[1], Markus D. Petters[2], Allan K. Bertram[3], Yutaka Tobo[4,5], Ryan H. Mason[3], Kaitlyn J. Suski[1,6], Christina S. McCluskey[1], Ezra J. T. Levin[1], Gregory P. Schill[1], Yvonne Boose[7], Anne Marie Rauker[1], Anna J. Miller[8], Jake Zaragoza[1,9], Katherine Rocci[10], Nicholas E. Rothfuss[2], Hans P. Taylor[2], John D. Hader[2], Cedric Chou[3], J. Alex Huffman[11], Ulrich Pöschl[12], Anthony J. Prenni[13], and Sonia M. Kreidenweis[1]

[1]Department of Atmospheric Science, Colorado State University, Fort Collins, CO, 80523, USA
    [2]Department of Marine, Earth and Atmospheric Sciences, North Carolina State University, Raleigh, NC, 27695 USA
    [3]Department of Chemistry, University of British Columbia, Vancouver, BC, V6T1Z1, Canada
    [4]National Institute of Polar Research, 10-3 Midori-cho, Tachikawa, Tokyo, 90-8518, Japan
[5]Department of Polar Science, School of Multidisciplinary Sciences, SOKENDAI (The Graduate School for Advanced Studies), Tachikawa, Tokyo, 90-8518, Japan
    [6]Now at Pacific Northwest National Laboratory, Richland, WA, 99352, USA
    [7]Karlsruhe Institute of Technology, Institute of Meteorology and Climate Research (IMK-IFU), 82467 Garmisch-Partenkirchen, Germany
[8]Department of Chemistry, Reed College, Portland, OR, 97202, USA
    [9]Now at Air Resource Specialists, Fort Collins, CO, 80525, USA
    [10]Department of Earth Sciences, University of New Hampshire, Durham, NH, 03824, USA
    [11]Department of Chemistry & Biochemistry, University of Denver, Denver, CO, 80210, USA
    [12]Department of Multiphase Chemistry, Max Planck Institute for Chemistry, D-55128, Mainz, Germany
[13]National Park Service, Air Resources Division, Lakewood, CO, 80228, USA

*Correspondence to*: Paul J. DeMott (Paul.Demott@Colostate.edu)

**Abstract.** A number of new measurement methods for ice nucleating particles (INPs) have been introduced in recent years, and it is important to address how these methods compare. Laboratory comparisons of instruments sampling major INP types are common, but few comparisons have occurred for ambient aerosol measurements exploring the

utility, consistency and complementarity of different methods to cover the large dynamic range of INP concentrations that exists in the atmosphere. In this study, we assess the comparability of four offline immersion freezing measurement methods (Colorado State University Ice Spectrometer, IS; North Carolina State University Cold Stage, CS; National Institute for Polar Research Cryogenic Refrigerator Applied to Freezing Test, CRAFT; University of British Columbia Micro-Orifice Uniform Deposit Impactor – Droplet Freezing Technique, MOUDI-

DFT) and an online method (continuous flow diffusion chamber, CFDC) used in a manner deemed to promote/maximize immersion freezing, for the detection of INP in ambient aerosols at different locations and in different sampling scenarios. We also investigated the comparability of different aerosol collection methods used with offline immersion freezing instruments. Excellent agreement between all methods could be obtained for several cases of co-sampling with perfect temporal overlap. Even for sampling periods that were not fully equivalent, the

deviations between atmospheric INP number concentrations measured with different methods were mostly less than





one order of magnitude. In some cases, however, the deviations were larger and not explicable without sampling and measurement artifacts. Overall, the immersion freezing methods seem to effectively capture INP that activate as single particles in the modestly supercooled temperature regime (>-20°C), although more comparisons are needed in this temperature regime that is difficult to capture with online methods. Relative to the CFDC method, three

immersion freezing methods that disperse particles into a bulk liquid (IS, CS, CRAFT) exhibit a positive bias in measured INP number concentrations at below -20ºC, increasing with decreasing temperature. This bias was present, but much less pronounced for a method that condenses separate water droplets onto limited numbers of particles prior to cooling and freezing (MOUDI-DFT). Potential reasons for the observed differences are discussed, and further investigations are required to elucidate the role of all factors involved.

## 50  1 Introduction

Heterogeneous ice nucleation by atmospheric aerosols impacts the microphysical composition, radiative properties and precipitation processes in clouds colder than 0˚C. These interactions are complex, and any first assessment of the role of different particles on ice formation, cloud properties and climate requires more observations of ice nucleating particles (INPs, as defined by Vali et al., 2015) present in ambient air. To quantify the initial stage of ice

nucleation in the atmosphere, multiple sampling techniques are now being used in field studies (Hader et al., 2014; Mason et al., 2015; DeMott et al., 2015; Stopelli et al, 2015; Boose et al., 2016; Schrod et al., 2016a,b). Since these various measurements are being used as bases for developing numerical model parameterizations for different emission sources, their comparability should be assessed. In this study, we focus on ice nucleation measurements in the mixed-phase cloud temperature regime (0 to -38°C), where heterogeneous ice nucleation is the only trigger for

primary ice initiation. Within this regime, INP number concentration can increase up to 10 orders of magnitude as temperatures cool from -5 to -35°C (DeMott et al., 2015; DeMott et al., 2016; Hiranuma et al., 2015; Murray et al., 2012; Petters and Wright, 2015), and there can be up to 2-3 orders of magnitude of temporal and spatial variability at a single temperature (by any given method) (DeMott et al. 2010; Petters and Wright, 2015).

This study compares results from an online INP measurement method used over the last 25 years, the Colorado

State University (CSU) continuous flow diffusion chamber (CFDC), with four offline immersion freezing methods for INP measurements. These four variants immerse particles into variously-sized liquid volumes/droplets which are cooled to freezing in different ways in order to measure the immersion freezing INP number per volume of air. In this study, comparisons are made only for times when the CFDC instrument operated in a manner which emphasized immersion freezing contributions to ice nucleation (DeMott et al., 2015). A principal reason to evaluate consistency

between approaches, and in ambient air, is because offline methods collect large enough sample volumes to estimate INP number concentrations active at modest supercooling (as warm as -5°C), where online instruments are unable to obtain statistically significant data samples; comparability between off- and online methods can be assessed in temperature regions of overlap. Another reason for such a comparison is to gauge the magnitude of uncertainties when only a single INP measurement method is used or when data sets from different instruments are combined

toward addressing a scientific question. This study differs from previous efforts in that comparisons have been





restricted to the ambient atmosphere, where presumably the compositions of INPs are more diverse and likely different than for single INP types often examined in laboratory studies. In one set of laboratory studies (Hiranuma et al., 2015), discrepancies between online and offline methods were noted for sampling NX-illite INPs. In particular, bulk, offline freezing methods estimated INP ice nucleation efficiencies that were 10 to 1000 times lower

than found with continuous flow chambers and the AIDA (Aerosol Interaction and Dynamics in the Atmosphere) expansion cloud chamber for temperatures warmer than about -25°C. Similar discrepancies were discussed by Emersic et al. (2016). Impacts of dry dispersion versus wet immersion on the agglomeration properties and the exposures of active sites were implicated in varied ways in both studies for explaining discrepancies. Grawe et al. (2016) also noted discrepancies occurring in single particle activation via immersion freezing in the LACIS (Leipzig

Aerosol Cloud Interaction Simulator) instrument for certain, but not all, combustion ash particles. In contrast, no discrepancies were reported in processing wet dispersed ice nucleating bacteria from Snomax® (Wex et al., 2014). Nevertheless, many of the reported laboratory results have thus far focused on a specific INP type that was shared across laboratories and for which individual investigators were allowed to determine protocols for generation as an aerosol or production of liquid suspensions for the different methods used. Here, by contrast, we focus on co-located

sampling of ambient aerosol, for which no more than two methods have hitherto been used in a single published study using this approach.

The goal of this inter-comparison is to assess the status and potential for using single or combinations of INP measurement methods to access and measure the dynamic range of atmospheric INP concentrations active for ice initiation in mixed-phase clouds. The assessment assumes that the time-dependence is subordinate to the

temperature dependence of the freezing nucleation process. The scientific basis of this assumption and its implications for the assessment are discussed. We address the magnitude of agreement, how particle collection methods may influence immersion freezing measurements, and whether obvious biases appear, for example due to the different size ranges of particles that may be collected in offline and online measurement systems. This study is not intended as a comprehensive evaluation, but rather a first assessment using some of the most common methods

likely to be applied for atmospheric sampling in the coming years.

## 2 Methods

Several INP measurement methods, most with a legacy of previous atmospheric measurements, are herein inter-compared during sampling of ambient aerosols. This section describes the instruments, details of sampling protocol and processing, and sampling sites.

### 2.1 INP measurement systems

### 2.1.1 Colorado State University CFDCs

Online INP measurements were made with two CSU CFDCs, designed for mobile and aircraft deployments, but otherwise identical (Eidhammer et al., 2010; DeMott et al., 2015). As described in these previous publications, aerosol flows vertically downward in a central lamina between concentric, cylindrical walls that are ice coated and





thermally controlled at different temperatures. Setting a temperature difference between the colder (inner) and warmer (outer) ice walls in the upper "growth" region establishes a nearly steady-state relative humidity where ice nucleation and ice crystal growth can occur over a few seconds. The temperatures of the inner and outer walls are set to the same value in the lower "evaporation" region of the chamber, which promotes evaporation of water droplets and wet aerosols, but retains activated ice particles at larger sizes that can be detected as optically-distinct for

counting as INPs with an optical particle counter. For this study, the aerosol lamina was 15% of the total volumetric flow of 10 L min$^{-1}$. Filtered and dried air was recirculated as sheath flow (8.5 L min$^{-1}$). Also for this study, a nominal water-supersaturated condition of 105% RH was chosen for operation at all temperatures. This selection was made to force activation of cloud droplets on aerosols at temperatures where some proportion could freeze during the transit time in the instruments, allowing for the most direct comparison possible to the offline immersion freezing

methods. Previous studies have explored the need to set the RH in CFDC style instruments to values far above that expected in natural clouds (100-101% RH) in order to mimic this freezing process (Petters et al., 2009; DeMott et al., 2010; 2015). Although DeMott et al. (2015) showed in laboratory studies that operational RH up to 109% might be required for full expression of freezing in the CFDC, 105% is the value that has been consistently used in field studies so that liquid droplets do not survive through the evaporation region and be counted as false positive INPs.

For mineral dusts, at least, operation at 105% could miss up to a factor of 3 (DeMott et al., 2015) INP number concentrations that ultimately activate via immersion freezing or some combination of nucleation mechanisms. It is unknown if this factor exists for all INP types. Hence, no correction factor was applied to the CFDC data here, but the implications of the factor of 3 will be discussed.

Aerosol particles at sizes that might confound optical detection of (i.e., be mistakenly counted as) ice crystals

were removed upstream of the CFDC using dual single-jet impactors set to a cut-point aerodynamic diameter of 2.4 μm. This creates a sampling bias for the CFDC versus other systems that capture larger particles for immersion freezing experiments, but is required to assure detection of activated ice crystals that typically exit the CFDC at optical diameters approximately >4 μm.

Interval periods of sampling filtered air within the overall sampling period were used to correct for any

background frost influences on INP counts. Error bars in average INP number concentrations are given by twice the Poisson sampling error for counting INPs. Particle losses in upstream tubing, the aerosol impactor, and the inlet manifold of the CFDC have previously been estimated as 10% for particles with diameter 0.1 to 0.8 μm (Prenni et al., 2009), and we apply this correction to data for this paper.

The typical lower CFDC detection limit for integrated sampling periods on the order of 10 min is~ 0.2 L$^{-1}$,

although application of a statistical significance test defining 95% confidence intervals has demonstrated that INP concentrations sometimes need to exceed values as high as 3 L$^{-1}$ to be considered statistically significant (Suski et al., 2017). Consequently, as a special sampling aide in these studies, an aerosol concentrator (Model 4240, MSP Corporation) was used upstream of the CFDC in some cases to enhance INP number concentrations and facilitate statistically significant quantification of INP number concentrations below the typical limit of detection. The

enhancement of aerosol concentrations using this dual virtual impactor method affects only particles of diameter >0.5 μm and varies from a factor of 10 at this diameter up to a factor of about 140 at sizes above 1 μm (Tobo et al.,





2013). The concentration factor achieved for ambient INPs then depends on the INP size distribution, which is difficult to know a priori. The methods outlined in Tobo et al. (2013) were followed to define the concentration factor, using the ratio of CFDC INP number concentrations with and without the concentrator under conditions

where statistical significance of measurement was achieved without the concentrator. This was assessed over the term of measurements for each site in the study, and applied to all CFDC data when using the aerosol concentrator. An example of measurements on and off of the concentrator for one of the sampling periods used in this study is shown in the Supplement, Fig. S1. Use of the aerosol concentrator is indicated in individual cases in the data tables, also included in the Supplement.

**2.1.2 North Carolina State University CS**

The design of the North Carolina State University (NC State) cold stage-supported droplet freezing assay (CS) and data reduction methods are described in detail in Wright and Petters (2013) and Hader et al. (2014).

Droplet populations of three distinct droplet size ranges may be investigated in the CS; these are termed pico-, nano-, and microdrops. Pico-drops are generated by mixing a 15 μL aliquot of bulk suspension (particles placed into

liquid by methods outlined below) with squalene and emulsifying the hydrocarbon-water mixture using a vortex mixer. The emulsion is poured into the CS sample tray, consisting of an aluminum dish holding a hydrophobic glass slide. Approximately 400 to 800 droplets with a typical diameter of ~85 μm are analyzed in this manner for each collected sample. Nanodrops are generated by manually placing drops with a syringe needle tip on a squalene covered glass slide and letting the drops settle to the squalene-glass interface. Approximately 80 droplets are

typically analyzed per experiment with a typical diameter of ~660 μm. Microdrops are placed directly on the hydrophobic glass slide using an electronic micropipette. In contrast to the pico- and nanodrops, these drops are in contact with gas-phase composed of dry $N_2$. Up to 256 drops of diameter ~1240 μm (1μl) can be investigated in a single experiment. For all experiments, the CS was cooled at a constant rate of 1ºC min$^{-1}$ and the number of unfrozen drops was recorded using a microscope in increments of $dT = 0.17$ºC resolution. To account for slightly

higher temperatures of the squalene relative to the glass slide, a temperature calibration was applied to the drop freezing data (Hader et al., 2014). The resulting data were inverted to find the cumulative concentration of INPs ($C_{INPs}(T)$) per volume of liquid at temperature, T, using the method of Vali (1971),

$$C_{INPs}(T) = -\left(\frac{1}{V}\right) \ln\left(\frac{N_u(T)}{N}\right) \tag{1}$$

where $N_u$ is the unfrozen number of an initial N liquid entities (droplets in this case) of volume $V$. Conversion to

number concentration of INPs per volume of air ($n_{INPs}(T)$) is determined by,

$$n_{INPs}(T) = C_{INPs}(T)\left(\frac{V_w}{V_s}\right) \tag{2}$$

where $V_w$ is the volume of liquid suspension (same units as used to compute $C_{INPs}(T)$) and $V_s$ is the sample volume (L) of air collected.

To minimize sample heterogeneity, only droplets with 78 μm $< D_p <$ 102 μm were included in the calculation

of $n_{INPs}$ (T) for picodrops. No restriction was applied to the nanodrops or microdrops. Furthermore, the warmest two percent of data was removed after the calculation of $C_{INPs}$ (T) but before plotting for the pico- and nanodrops due to





large uncertainty stemming from poor counting statistics (Hader et al., 2014). The INP content of the ultrapure water (see section 2.2) was measured in the above manner between -20ºC and -35ºC. The effective INP content was determined by subtracting the background INP numbers from the ultrapure water from observed $n_{INP}$ (T). No

impurities were detected at $T > -20$ºC.

### 2.1.3 University of British Columbia MOUDI-DFT

The second immersion freezing method involved freezing of droplets grown on substrate-collected particles in a temperature and humidity controlled flow cell (Mason et al., 2015) and is referred to as the droplet freezing technique (DFT). A micro-orifice uniform deposit impactor (MOUDI; MSP Corp.) was used to size-select particles

from known volumes of air onto a substrate for direct DFT analysis in a number of cases (MOUDI-DFT, Mason et al., 2015). The MOUDI collected size-selected particles onto multiple hydrophobic glass cover slips (HR3-215; Hampton Research). For the measurements performed in Kansas, United States, stages 2-9 of the MOUDI were used corresponding to particle size bins of 10–5.6, 5.6–3.2, 3.2–1.8, 1.8–1.0, 1.0–0.56, 0.56–0.32, 0.32–0.18, and 0.18–0.10 μm (50% cutoff aerodynamic diameter; Marple et al., 1991), respectively. For the measurements at CSU,

stages 2-8 were used and for the measurements at Manitou (Colorado) Experimental Forest, stages 2–7 were used. Samples were vacuum-sealed and stored at 4°C in a fridge (or using cold packs if shipment of the samples was required) until cold stage flow cell measurements were performed at the University of British Columbia.

For DFT analysis, droplets were grown in the flow cell by decreasing temperature to 0°C and passing a humidified flow of He gas over the slides. Water was allowed to condense until approximately 100 μm diameter water droplets

formed on the collected particles, typically covering several to some tens of particles depending on loading. Droplets were then monitored for freezing using a coupled optical microscope (Axiolab; Zeiss, Oberkochen, Germany) with a 5× magnification objective, as temperature was lowered at a constant rate. A CCD camera connected to the optical microscope recorded a digital video while a resistance temperature detector recorded the temperature. A cooling rate of 10ºC min$^{-1}$ (from 0°C to -40°C) was used in these studies to either minimize freezing of droplets due to contact of

a growing crystal or minimize evaporation of unfrozen droplets due to the Bergeron-Findeisen process, i.e. growth of the existing ice crystals at the expense of the surrounding liquid droplets (Mason et al., 2015). The liquid droplet may evaporate or the frozen droplet will grow towards and eventually contact a liquid droplet, causing it to freeze. If a droplet is lost to evaporation or to non-immersion freezing, two assumptions are made:

1) That the droplet contained an INP and would have frozen by immersion (on its own) at the same temperature as

the non-immersion/evaporation event. This gives an upper limit to the calculated INP concentration

2) That the droplet contained no INP and would not have frozen until homogeneous temperatures, which are around -36ºC in the flow cell used. This assumption provides a lower limit to the calculated INP concentration at a given T. The method to obtain the INP number concentrations in air follows a similar basis as for the CS, but with modest differences as,

$$n_{INPs}(T) = -\ln\left(\frac{N_u(T)}{N}\right) N \left(\frac{A_{\text{deposit}}}{A_{\text{DFT}}V_s}\right) f_{\text{nu}} f_{\text{ne}} \tag{3}$$

where $N$ is the total number of droplets condensed onto the sample in this case, $A_{\text{deposit}}$ is the total area of the sample deposit on the hydrophobic glass cover slip, $A_{\text{DFT}}$ is the area of the sample monitored in the digital video during the



droplet freezing experiment, $V_s$ is the volume of air sampled by the MOUDI, $f_{ne}$ is a correction factor to account for
the uncertainty associated with the number of nucleation events in each experiment, and $f_{nu}$ is a correction factor to
account for non-uniformity in particle concentration across each MOUDI sample (Mason et al., 2015; Mason et al.,
2016).

### 2.1.4 Colorado State University IS

The CSU ice spectrometer (IS) (Hill et al., 2014; Hill et al., 2016; Hiranuma et al., 2015), measures freezing in an
array of liquid aliquots held in a temperature-controlled block. For IS processing, aerosol particles in suspensions
are distributed into 24 to 48 aliquots of 40-80 µL held in sterile 96-well PCR trays (µCycler, Life Science Products).
The numbers of wells frozen are counted at 0.5 or 1°C intervals during cooling at a rate of 0.33°C min$^{-1}$. Calculation
of $n_{INPs}$ (T) was made using Eqs. (1) and (2), where V was the aliquot volume. Control wells of ultrapure water (see
section 2.2) were also cooled, and correction for any frozen aliquots in the pure water control versus temperature
was made in all cases, similar to the CS method. Uncertainties are given as binomial sampling confidence intervals
230    (95%).

### 2.1.5 National Institute of Polar Research CRAFT

The Cryogenic Refrigerator Applied to Freezing Test (CRAFT) device has been described in detail by Tobo (2016).
CRAFT is a classical cold plate device akin to the DFT and the CS instruments, but involves procedures to assure
sample isolation, primarily from the cold plate surface using a layer of Vaseline oil. Droplets containing collected
aerosols are pipetted in a clean hood onto the coated aluminum plate that is then set on the stage of a portable
Stirling-engine-based refrigeration device (CRYO PORTER, Model CS-80CP, Scinics Corporation). The freezing
device is also operated in a booth that is aspirated with clean air. The temperature of the plate was measured using a
single temperature sensor, and the uncertainty of temperature is 0.2°C.

For each CRAFT measurement, 49 droplets with a volume of 5 µL were used and the temperature was lowered
at a rate of 1°C min$^{-1}$ until all the droplets froze. Results of control experiments with pure water droplets were used
to correct for any contamination introduced by water. Each freezing experiment was monitored by a conventional
video camera. Video image analysis was used to establish the number fractions of droplets frozen and unfrozen at
0.5°C intervals. Analyses of $n_{INPs}$ (T) followed the same scheme as used for the CS and IS measurements.

### 2.2 Aerosol collection methods and processing for immersion freezing studies

At different times, ambient aerosol samples were collected directly into liquid or onto filters, for subsequent
resuspension into liquid. Collection directly into liquid was done using a glass Bioaerosol sampler (SKC Inc.),
hereafter termed the BioSampler. This unit was typically placed on a table at 1.2 m above ground level. The
BioSampler directs particles into a sample cup filled with 20 mL of ultrapure water (18.2 MΩ cm resistivity and
0.02 µm filtered using an Anotop syringe filter (Whatman, GE Healthcare Life Sciences)) where they impinge to
form an aqueous suspension. Particle collection efficiencies for this technique exceed 80% for particles larger than
200 nm and approach 100% for particles larger than 1 µm (Willeke et al.,1998). Particles with diameter $D_p$ > 10 µm





are expected to impact the inlet wall (Hader et al., 2014). Sample flow rate was 12.5 L min$^{-1}$, and impaction liquid was replenished every 20-30 min by adding ultrapure water into the collection cup.

For IS-only and some shared samples, particles were also collected onto pre-sterilized 47 mm diameter
Nuclepore$^{TM}$ track-etched polycarbonate membranes (Whatman, GE Healthcare Life Sciences). Filters were pre-cleaned by soaking in 10% $H_2O_2$ for 10 min, followed by three rinses in ultrapure water, and were dried on foil in a particle-free, laminar flow cabinet. Filters were held open-faced in sterile Nalgene filter units (Thermo Scientific, Rochester, NY). Flow rates varied from about 8 to 13 L min$^{-1}$ for ambient temperature and pressure conditions in different studies. Collection onto 0.2 µm pore-diameter filters was typical, although comparison versus 3 µm pore-
diameter filters was also done in some initial experiments. Both filter types were of ~10 µm average thickness and 15% porosity. On the basis of theoretical collection efficiencies (Spurny and Lodge, 1972), the 0.2 µm pore filters should have collected particles of all sizes with very high efficiency, the lowest efficiency being at about 0.1 µm (~80%). In contrast, the filters with 3 µm pores are expected to collect 15% and 55% of all particles at sizes of 0.4 and 1 µm, respectively, increasing to >75% collection at sizes above 1.5 µm. In this manner, the larger pore size
emphasizes the contributions of supermicron aerosols to immersion freezing INPs.

After particle collection, filters were stored frozen at -25 or -80°C in sealed sterile petri dishes until they could be processed (few hours to few months). Selected tests processing samples immediately versus after frozen storage (not shown) indicated no impact of this storage.

For processing of INP freezing spectra, filters were transferred to sterile, 50 mL Falcon polypropylene tubes
(Corning Life Sciences), immersed in 7.0-10.0 mL of ultrapure water, and tumbled for 30 min in a rotator (Roto-Torque, Cole-Palmer) to suspend particles in liquid. Common liquid suspensions were shared amongst methods in some cases (see section 2.3), following freezing and shipping to different investigators. We detected no measurable impact of processing rinsed suspensions immediately versus after freezing of the bulk water, mostly supported by other recent studies (Beall et al., 2017). We will note that while all immersion freezing methods performed tests
comparing freezing of the liquid samples and the purified water used in their setups, and corrected for pure water freezing events, no correction is made for any INPs that might be released from the filters used for collection. We have found that filters release a modest number of INPs active at lower temperatures, even after the pre-treatments with $H_2O_2$ and purified water. A detailed analysis of this will be presented in a future publication. The percentages of undiluted INPs due to such contamination is ~3 % in the -25 to -30ºC range, and since immersion freezing
measurements at these temperatures require dilution of liquid samples by 100 to 3000 times, we neglected any corrections.

## 2.3 Sampling sites/periods and objectives

Sampling sites represent a variety of ecosystems, climates and elevations across the Western U.S., including agricultural regions of the U.S. High Plains, intermountain desert regions, and a coastal site. The majority of data
included in this inter-comparison involved periods that did not include all groups and were not temporally-aligned for all instrument systems. Nevertheless, substantial overlap of sampling periods occurred in all cases. Very often, the CFDC sampling was conducted to obtain data at multiple temperatures, while offline collections were made for



longer periods to obtain integrated INP temperature spectra. Times when the sampling periods were the same for the offline systems and for the CFDC, while it was operating at a single temperature, are listed in Table 1. Other site

locations, characteristics, and instruments participating when there were overlapping sample periods are listed in Table 2.

### 2.3.1 Colorado State University, Fort Collins, CO, USA

Sampling was conducted outside of the Atmospheric Chemistry building at Colorado State University at different times and including different methods. The laboratory site is on a small hill on the western edge of the Fort Collins

urban area, residing amongst surrounding grasslands. Initially, a series of measurement days were conducted in which collections for three immersion freezing methods were made while the CFDC sampled at a single temperature for the entire sampling period. While this protocol permitted only a single comparison point versus the temperature spectra obtained by offline measurements, the purpose was to obtain a statistically significant CFDC $n_{INPs}$ value during the course of time-integrated offline samples and to assure that any signal variance occurring during

sampling was the same for all measurements. Such aligned sampling was conducted on five different days (see Table 1). Participating in these temporally-aligned experiments were the IS, CS, and MOUDI-DFT instruments. For these periods, the filter sampling units, BioSampler and (when used) MOUDI sampling units were set in close proximity and at the same sampling elevation. Filter suspensions from the two pore-size (0.2 and 3.0 μm) filter collections and from the BioSampler were shared for IS and CS measurements. All CS data were analyzed using the

pico- and nanodrop technique.

Sampling was also conducted at CSU without exact temporal overlap of CFDC, IS, and CRAFT method measurements, as noted in Table 2. CRAFT filters (0.2 μm pore size) were drawn for 6 hours at a flow rate of 10 L min$^{-1}$ at standard temperature and pressure (STP) conditions (T = 273 K, 1013.5 mb). IS filter (0.2 μm pore size) were drawn for 4 hours at a flow rate of 13 L min$^{-1}$ at ambient temperature and pressure. The CFDC sample was

temporally aligned with the IS sample, and single operating temperatures were used.

### 2.3.2 Northern Colorado, USA, agricultural region

Sampling over previously harvested fields during Fall 2010 was conducted at a rural site approximately 26 km NNE of the CSU Atmospheric Chemistry building, at Grant Family Farms, near the village of Waverly, CO. The sampling field sites on different days, sampling protocol and the results used in the present study, are discussed in detail by

Garcia et al. (2012). Sampling by CFDC and IS (BioSampler) were temporally overlapped in this study. This site is referred to as NoCO in the data tables in the Supplement.

### 2.3.3 Manitou Experimental Forest, CO, USA

Sampling within an open forest site (Manitou Experimental Forest Observatory, hereafter MEFO) as part of the Bio-hydro-atmosphere interactions of Energy, Aerosols, Carbon, H$_2$O, Organics & Nitrogen project (Ortega et al., 2014)

during Summer 2011 was conducted as described by Huffman et al. (2013), Prenni et al. (2013) and Tobo et al.





(2013). Only two selected periods from that study for which there was partial overlap of samples from the CFDC and MOUDI-DFT methods were available for this study.

### 2.3.4 Kansas, USA, agricultural region

Sampling periods were conducted in and around the times of different crop harvesting at Kansas State University
Northwest Research Extension Center in Colby, KS as part of a larger study (Suski et al., 2017). Sampling periods used for this study were during mornings before or evenings following harvesting of various crops, and during daytime near fields being harvested of soy and sorghum crops. CFDC sampling was conducted from the CSU Mobile Laboratory facility, using gasoline powered generators, as described previously by McCluskey et al. (2014). The mobile laboratory was in all cases well upwind of the generators. Aerosols were sampled through an inlet
comprised of a stainless-steel rain hat with a ½" OD stainless steel tube attached. MOUDI-DFT (Mason et al., 2016) and filter samples were collected with their inlets at the same approximate elevation as the CFDC inlet, and used separate pumps for drawing samples. The CFDC scanned different temperatures during the IS filter (0.2 μm) and MOUDI-DFT sampling periods.

### 2.3.5 Southern Great Plains (SGP), USA, site

The site at Lamont, OK (Table 2) is the central instrumentation suite location for the U. S. Department of Energy's Atmospheric Radiation Measurement program, Southern Great Plains (SGP) field site. CFDC and IS instruments both drew air from a platform at 10 m above ground elevation at this site. Sampling occurred in a transition from dry to wet conditions in the Spring of 2014. The CFDC was operated to scan temperatures during the IS filter (0.2 μm) sampling period. A selection of representative days of data were chosen, and full study data will be included in a
separate publication.

### 2.3.6 Bodega Marine Laboratory, CA, USA

Sampling near Bodega Bay, CA (BBY in subsequent figures) occurred during the CalWater-2015 study (Ralph et al., 2015; Martin et al., 2016). The sampling site was at the University of California, Davis Bodega Marine Laboratory, ~100 m ENE of the seashore and ~30 m north of the northernmost permanent building at the site
(Martin et al., 2016). The CFDC and IS instruments sampled from approximately 4 m above the surface. The CFDC was operated to scan temperatures during the IS filter (0.2 μm) sampling period. CS BioSampler samples, overlapping with IS and CFDC sampling, were drawn from an elevation of 1 m above the vegetated surface, approximately 20 m west of the other samplers.  All BBY CS data are analyzed using the microdrop technique. A few representative days are chosen from the data set for comparison of IS and CS data with CFDC data. Comparison
of the complete CS and IS data sets will be included in a publication in preparation.

### 2.3.7 Canyonlands Research Center, UT, USA

The Nature Conservancy's Canyonlands Research Center is an intermountain (Rocky Mountains, U.S.), high desert site located adjacent to Canyonlands National Park in SE Utah. Sampling occurred in May of 2016. IS and CRAFT



filters were drawn at 1.2 m above ground, the same elevation as the CFDC inlet. CRAFT filters were drawn for 6

hours at a flow rate of 10 L min-1 at standard temperature and pressure (STP) conditions (T = 273 K, 1013.5 mb), at

this site and at CSU. IS filters (0.2 μm pore size) were drawn for 4 hours at a flow rate of 13 L min$^{-1}$. CFDC

sampling overlapped with the IS filter period, but operating temperature was varied.

**3 Results**

**3.1 Comparison of cases with perfect temporal overlap of sample data collections**

Figure 1a compares IS and CFDC data for two 4-hour study periods at the CSU site. In the figure CFDC INP

concentrations at -16ºC are integrated and averaged for the entire IS filter sampling period for comparison to IS data

collected both on filters and using the BioSampler. The lack of significant difference in IS $n_{INPs}$ measured with the

filters of 0.2 and 3 μm pore sizes implies that most INPs were likely large enough to be captured effectively by both

filter types, and hence the INP mode size is likely 1 μm or larger.  This is consistent with a size that is collected with

high efficiency in the Biosampler, for which similar INP concentrations were measured. This example also shows

the uncertainties in temperature spectra of INP number concentration from the IS.  In this case, one can see a range

of about a 4 factor in INP number concentration and an equivalent range of 2 to 4ºC using different collection

methods, and in consideration of confidence in measurements made at any particular temperature.  The CFDC data

collected using the aerosol concentrator are in agreement within uncertainties of all particle collection methods in

this case.

In Fig. 1b, results are shown from a case where filter rinse suspension and BioSampler suspension were also

shared with the CS instrument for offline processing of samples collected from the CSU site on September 6, 2013.

There is significant overlap between the IS and CS data in the temperature range from -6 to -23°C (the lowest

temperature limit of IS processing for these particular experiments). No significant bias is discernable between IS

and CS data for any of the collection methods. Once again, correspondence of the CFDC data (using the aerosol

concentrator in this case) with other methods is good at a processing temperature of -18.2°C. However, the CFDC

data falls a factor of 2-5 lower than the immersion freezing methods. This is similar to data reported in Garcia et al.

(2012) for which the discrepancy was attributed primarily to the failure of the CFDC instrument to sample larger

aerosols. Nevertheless, results from this sampling day support the conclusions of general agreement between

methods obtained in Fig. 1a.

Figure 2 shows results from three additional cases for which there was perfect temporal co-sampling by the

CFDC, IS, CS and MOUDI-DFT methods. In these cases, the IS and CS shared samples of particles collected during

the same time period, while the MOUDI-DFT was operated independently. We note that the error bars on MOUDI

data reflect upper and lower bound estimates, as discussed in section 2.1.3. Figure 2 highlights some points already

made, but also the occurrence of a range of discrepancies in $n_{INPs}$ between the MOUDI-DFT and other methods, and

for CFDC data collected simultaneously at temperatures below -20°C. The CS method typically measures the

highest $n_{INPs}$ overall for the same collections of aerosols (filter or BioSampler), suggesting a temperature offset of at

least 1°C between these systems that may have as its source the temperature measurement of the liquid wells or





drops. The MOUDI-DFT results trend with the other immersion freezing methods on all days, but agree
quantitatively with them on only one of three days (Fig. 2a) and fall lower than $n_{INPs}$ determined by the CS and IS on
two other days; by a factor of 2 to 5 (Fig. 2c) in one case and 20 to 50 in the other (Fig. 2b). These two cases have
been discussed previously in Mason et al. (2015), and we will revisit the largest discrepancies in both cases in later
discussion. Similar to the MOUDI-DFT results, the CFDC data also show a consistent underestimate of $n_{INPs}$
compared to the CS and IS in all three cases, with a trend that increases from a factor of 2-4 at -23°C, up to 10 times
at -30°C (Fig. 2a).

### 3.2 Comparison for cases of imperfect temporal overlap of sample data collections

The data shown in Fig. 1 and Fig. 2, for which there was complete temporal overlap of observations, provide a
limited number of evaluations of measurement correspondence and uncertainties that may occur due to different size
ranges of collection and variabilities that may occur over varied sampling times as measured across the mixed-phase
cloud temperature regime. This situation will surely be improved in future studies as many different instrument
teams worldwide begin to compare data collected at common sites. To expand understanding, we considered all
cases in which the CFDC was sampling simultaneously with other methods, but without the restriction of a single
CFDC processing temperature for the full sampling period. There are also cases when the offline sample periods
overlapped but did not perfectly align.  Thus, while seeking further insights by folding in data from additional times
and collection sites, we must acknowledge that such comparisons leave open the possibility that temporal variability
may impact comparison of methods. Nevertheless, this replicates many field study situations where multiple ice
nucleating instruments may be deployed, but may not sample for the same time periods.

In Fig. 3, we combine periods of perfect sampling overlap with these other cases for which one or more of the
immersion freezing methods were performed for a few-hour period, during which CFDC sampling intervals
(typically 10-15 minutes at a single temperature) occurred. Comparison of the CFDC and IS measurements are
shown in Fig. 3a. These results reinforce those in Fig. 2, indicating that the IS assessment of $n_{INPs}$ agrees on average
with the CFDC-measured values when the CFDC processed particles at 105% RH at the lower end of the dynamic
range of $n_{INPs}$. The IS method, however, measures higher concentrations than the CFDC at higher $n_{INPs}$, resulting in a
non-unity relational slope. Higher $n_{INPs}$ typically occur with decreasing temperatures, as will be reiterated in later
discussion. These results are similar regardless of measurement site, but with relatively high variability in the
relation between single CFDC and IS measurements even at a single site, and with greater discrepancy in the data
set from Colby, KS, which we suggest is the result of an abundance of larger INPs not sampled by the CFDC during
this harvesting period.

The MOUDI-DFT data show the best correspondence overall versus the CFDC measurements (Fig. 3b),
irrespective of whether all aerosol sizes are considered for the DFT measurement or are limited to a range of particle
sizes similar to those entering the CFDC.  There is a slight positive bias for the MOUDI-DFT method when all sizes
are considered, as expected given the CFDC limitation on particle sizes sampled.



Overlapping comparisons between the CS and CFDC, and CRAFT and CFDC, while more limited (Fig. 3c), show a relatively high bias of the CS and CRAFT data, most exaggerated at higher $n_{INPs}$ and correlated with lower temperatures as discussed shortly.

Overall comparisons by offline method versus the CFDC are shown in Fig. 3d. These demonstrate that although a consistent linear (but not 1:1) relationship could be inferred between offline immersion freezing and CFDC measurements, discrepancies for all methods and sampling times taken together at a CFDC $n_{INP}$ of 1 L$^{-1}$ can reach nearly two orders of magnitude. Discrepancies appear to reduce to within about 1 order of magnitude at higher $n_{INPs}$, although the degree to which this is real or the result of a fewer number of cases is not yet clear. We may note of course that CFDC measurements have their greatest uncertainties in the range of concentrations at or below 1 L$^{-1}$.

The same data sets used in Fig. 3, and compiled in SI Table 1, are used in Fig. 4 to explore the temperature dependence of immersion freezing measurement results versus the CFDC when all sampling scenarios are considered (multiple aerosol scenarios, perfect or imperfect overlap of sampling times). In examining the IS versus CFDC comparisons (Fig. 4a), the scatter in the relation is again the most striking feature, while the temperature-dependent bias also becomes clear to a greater or lesser degree at all sampling sites, the least at CSU and the SGP site, and the most at Bodega Bay and in the harvesting period in Kansas. The strong positive bias of INP measurements by the IS at lower temperatures in Kansas is not consistent with the fact that larger INPs (>2.5 μm), that are not sampled by the CFDC, are not thought to dominate INPs at lower temperatures (Mason et al., 2016). A more modest positive temperature bias is noted in comparing MOUDI and CFDC concentrations versus temperature at below -25C (Fig. 4b), and the underestimate of INP concentrations due to the elimination of coarse mode aerosols in CFDC sampling ranges from about 2 to 4 times (see MOUDI "all" versus "size" in Fig. 4b), consistent with the estimates of coarse mode INP fractions by Mason et al. (2016). We may note similarly good agreement between INP concentrations measured by the CFDC and DFT methods across similar temperature ranges for marine aerosols (DeMott et al., 2016). Strong positive biases of CS and CRAFT measured INP concentrations versus the CFDC measurements are seen to progressively occur as temperatures decrease from -20 to -30°C (Fig. 4c).

## 4 Discussion

In this section, we summarize observations regarding comparisons of the INP measurement methods and discuss potential reasons for discrepancies that bear future investigation. It has been shown that there are times when multiple measurement techniques give excellent agreement for $n_{INPs}$ in the immersion freezing mode. Agreement is best at temperatures warmer than -20°C and for $n_{INPs}$ less than ~5 L$^{-1}$. At lower temperatures and higher $n_{INPs}$, most offline immersion freezing methods, with the exception of MOUDI-DFT, estimate higher $n_{INPs}$ than the online CFDC method, by ratios ranging from a few to 10 times. We must caution that the overall range of $n_{INPs}$ assessed and values present at different temperatures may reflect the aerosol measured at ground level at the selected sites and times, scenarios that may not represent all locations and times worldwide. Nor may these results be the same if the comparisons were made entirely for free tropospheric aerosols, for example as assessed from an aircraft or at





some mountaintop sites. Nevertheless, the potential issues in obtaining agreement between methods will be common in any sampling scenario.

A factor in any series of immersion freezing measurements is the time dependence of nucleation. In a study of the time-dependent freezing of kaolinite particles, Welti et al. (2012) demonstrated that the majority of freezing occurred within about a period of 10 s or less at the temperatures -30 to -37°C, with 0.8 μm diameter particles needing far less time for activation than 0.4 μm particles. Studies of freezing rates for other natural INP types across broader temperature ranges indicate that immersion freezing is indeed not a purely stochastic process and is far more sensitive to temperature, with the consequence that the increase in $n_{INPs}$ achieved by holding for hours at one temperature are typically overcome by a few degrees of additional cooling (Vali, 2014; Wright et al. 2013). The CFDC $n_{INPs}$ attributed here to immersion freezing were obtained for a total processing time of approximately 7 s, the last 2 s of which activated droplets are evaporating (DeMott et al., 2015). This residence time is constrained by flow rates required for limiting thermally-driven reverse flow circulations in the CFDC. By comparison to the Welti et al. (2012) study, it seems likely that the CFDC activation times allow for capturing the majority of immersion freezing activity in most circumstances. Nevertheless, we expect that the CFDC might underestimate $n_{INPs}$ to a greater extent than the IS measurements that are made while ramping at a very slow cooling rate equivalent to 1°C in 3 minutes. Since the DFT uses much faster cooling rates (5 - 10°C min$^{-1}$), this might explain the better correspondence with the CFDC data. However, it cannot explain the temperature-dependent nature of the bias between other immersion freezing methods and the CFDC, and so seems not the only source of this discrepancy.

Here we must also reiterate that the processing of submicron mode mineral dust particles at 105% RH in the CFDC has been shown to underestimate $n_{INPs}$ by an average, temperature-independent factor of 3 times, as confirmed by laboratory cloud chamber simulations. This factor was related to the fact that higher RH is typically required to fully activate all particles (hygroscopic or hydrophobic) as droplets to be available for freezing in the CFDC residence time (DeMott et al., 2015; Garimella et al., 2017). However, practical operation of the CFDC at higher RH (109% may be required for full activation) is prohibited in sampling of natural aerosol distributions because the largest aerosols could persist as droplets through the evaporation section of the instrument under these conditions, thus contaminating INP determination using optical sizing. Hence, it is unknown if natural INP populations are being underestimated for similar reasons. Based on the recent study of Garimella et al. (2017), it seems possible that underestimation of INP concentrations occurs for CFDC-style instruments independent of the aerosol type. Consequently, lines indicating a factor of 3 higher than the 1:1 relation have been placed on plots in the panels of Fig. 3. While it is noted that increasing the CFDC $n_{INPs}$ by 3 times leads to better overall agreement of CFDC data with the CS and CRAFT data especially, this constant expected offset does not explain the progressive underestimate of the CFDC in comparison to most immersion freezing methods (the IS, CS and CRAFT being most like other methods used worldwide) at higher $n_{INPs}$ and lower temperatures.

A factor that could artificially increase $n_{INPs}$ at lower temperatures in methods that immerse the entire aerosol population first into liquid (IS, CS, CRAFT) is the potential breakup of aggregates containing multiple INPs INPs (e.g., via the deflocculation of small aggregates as a result of the strong reduction in di- and trivalent cation concentrations in the deionized water used for making dilution series, or by the fragmentation of mucigels (Hill et




al., 2016)) and the possible dissolution release of surface-active INP materials present on single particles when
suspended in deionized water (O'Sullivan et al., 2016). It seems possible that such action would have the greatest
impact on INPs active at lower temperatures (rather than the most active INPs), since these may be small
clay/organic matter aggregates or flocs that fragment when exposed to deionized water. Since the MOUDI-DFT
method immerses a relatively small number of particles directly and without agitation in small drops prior to
freezing, it is interesting to note that the least temperature-dependent bias occurs for these measurements in
comparison to the CFDC. This point is shown more clearly by comparing only the offline immersion freezing
methods in Fig. 5. In this figure, the different measured INP concentrations are taken as a ratio versus the IS, which
sampled the most times and scenarios. Data at one degree temperature resolution are included in this comparison, as
compiled in Table S2. Again, relatively high variability of at least 1 order of magnitude at any temperature is noted
for the relations between methods. Among methods that involve immersion of all particles in a single volume of
water prior to setting up arrays (CS, CRAFT, IS), the IS falls to the low side of the other measurements by an
average factor of about 2.5. This is not a significant difference, in consideration of the likely temperature
uncertainties discussed in relation to Figures 1 and 2. The MOUDI-DFT method that immerses relatively small
populations of particles, shows relative equivalence to the full immersion methods at modest to moderately
supercooled temperatures, but measured consistently lower INP concentrations at below about -20°C in the few
cases when co-sampling was conducted with the IS (CSU and Kansas). This is consistent with the discrepancy seen
also versus CFDC data. Interestingly, a lower temperature enhancement of INPs appearing in full immersion
methods versus continuous flow methods was not observed in recent laboratory tests comparing many measurement
methods while sampling mineral, soil dust and biological particle samples that had been purposely limited to sizes
smaller than 2 μm (DeMott et al., 2017). While this points to coarse mode particles and their
dissolution/fragmentation into multiple INP units as the source of differences, future experiments will be needed to
confirm or deny that this is either an artifact or a behavior in natural aerosols that the CFDC cannot effectively
capture.

Particle size limitations lead to CFDC underestimates of $n_{INPs}$ in comparison to some immersion freezing
methods. This is because of the need to remove particles larger than 2.4 μm. This removal of larger aerosols is
necessary when differentiating grown ice crystals from aerosols by size alone. Even absent use of impactors, it
would be difficult for most online systems to effectively sample larger particles due to the design of inlet systems.
With reference to the study of Mason et al. (2016), which entailed sampling with the MOUDI-DFT method at
various sites, one might estimate that on average about 50% of INPs are at sizes larger than 2.4μm in the surface
boundary layer. Comparison of MOUDI-DFT with CFDC data in this study is consistent with this same estimate
(Fig. 3b). Again, this would not apparently explain a progressive slight increase in CFDC underestimation versus the
MOUDI-DFT at lower temperatures unless larger INPs specially dominate ice nucleation at lower temperatures, a
result not consistent with Mason et al. (2016).

In evaluating the low temperature discrepancies by noting the better correspondence of MOUDI-DFT and CFDC
methods, it is also necessary to note the potential issue of particle bounce in the MOUDI in some cases (Mason et
al., 2016). While the conditions for this to occur are not well quantified, since both INP size and phase state (as this





may be influenced by low relative humidity) may affect bounce, very dry conditions have been indicated as times when this may become an issue for MOUDI impaction onto the substrates used in the DFT instrument. Interestingly, average RH during the sampling period on November 13, 2013 (Fig. 2b) was between 15-20% during the sampling period versus 40-45% for days on either side (Figs. 2a, c), potentially impacting and explaining the MOUDI-DFT
results on this day. For this reason, this day was excluded in Fig. 3 to Fig. 5. Sample humidification of the system could mitigate this factor as a potential issue in future sampling.

## 5 Conclusions

This study has inspected the correspondence of ice nucleation measurement systems for co-sampling ambient ice nucleating particles. In this case, we considered systems for measuring immersion freezing nucleation with a
common online method used in a manner to induce activation of cloud droplets prior to ice nucleation. Excellent agreement was obtained under many conditions for samples that had perfect temporal overlap. In other cases, discrepancies can approach two orders of magnitude and are not explicable without inferring systematic artefacts inherent to one or more techniques. The results summarized in Fig. 3d show the uncertainties that can be expected when employing one or more of these instrument systems for measuring atmospheric INPs. Within these
uncertainties, the data suggest that the low bias of immersion freezing methods reported by Hiranuma et al. (2015) for sampling of individual surrogate dusts in the laboratory was not evident in these ambient data sets.

With regard to particle sampling methods for immersion freezing measurements, use of a BioSampler or a filter was interchangeable, at least for the continental boundary layer sampling for which these methods were compared. This was demonstrated for individual and for cross-technique methods (IS versus CS) for assessing immersion
freezing from the same samples. Since Nuclepore filters seem to efficiently capture and release INPs, these provide ease of use benefits in many field scenarios, although the role of retention of particles on some filter types has not been assessed here.

The strongest discrepancies between methods appear at both warmer and colder ends of the scale of mixed-phase cloud freezing temperatures. At the warmer end (T>-20°C), sampling statistics and uncertainties can dominate
comparisons of online and offline methods. Full explanations for the maximum 2 orders of magnitude range of variation in this temperature regime remains unresolved. In contrast, at lower temperatures the IS, CS and CRAFT methods measured more INPs than detected by the CFDC and MOUDI-DFT. Potential artifacts or biases are present in these comparisons and have been discussed here, including varied assessment of time dependence of ice nucleation, necessary exclusion of larger INPs by online instruments such as the CFDC, and immersion of all
particles into relatively large volumes of deionized water in most, but not all, immersion freezing methods versus activation of single particles in CFDCs. In addition, it is expected that all CFDC type instrument data may require correction for not being able to access full immersion of particles until higher RH than can commonly be used when sampling ambient particles, or else this issue requires future mitigation (e.g., insertion of particles into the instrument lamina could be improved). Hence, no assured conclusions regarding the sources of discrepancies can be
stated at this time except that size biases in sampling need to be acknowledged. Effort thus remains to make INP measurements fully quantitative and comparative across methods, if correspondence within less than one order of





magnitude is desired. Even amongst standard immersion freezing methods, uncertainties of a factor of a few $n_{INPs}$ and 2 to 4ºC are likely common on the basis of this study, and may be the best that can be achieved. Application of size selection to immersion freezing collections for comparison to MOUDI-DFT data (especially at lower
temperatures) and CFDC data, information on INP compositions inferred under all sampling scenarios to help constrain influences of various types (e.g., methods of Hill et al., 2016), and an inter-comparison of all methods versus a cloud parcel simulation chamber, considered as a *de facto* standard, would all assist resolution and improvement of understanding of measurement discrepancies.

*Data availability.* The data used are listed in the references, tables, Supplement, and in a digital repository at CSU
(https://dspace.library.colostate.edu).

*Competing interests.* The authors declare no competing interests.

*Acknowledgments.*  Funding for this work was provided by National Science foundation grants AGS1358495 (P. J. DeMott, T. C. J. Hill, K. J. Suski, and E. J. T. Levin), AGS1010851 (M. D. Petters and J. D. Hader) and AGS1450690 (M. D. Petters, N. Rothfuss, and H. Taylor), AGS1450760 (P. J. DeMott, T. C. J. Hill, C. S.
McCluskey, and S. M. Kreidenweis) and AGS1433517 (G. P. Schill). Support for operations at the U.S. Department of Energy Southern Great Plains site was provided by the Atmospheric Radiation Measurement (ARM) Climate Research Facility, a U.S. Department of Energy Office of Science user facility sponsored by the Office of Biological and Environmental Research. A. K. Bertram, R. H. Mason, and C. Chou acknowledge support of the Natural Sciences and Engineering Research Council of Canada. Y. Tobo acknowledges support from JSPS KAKENHI
Grant Number 15K13570, 16H06020, the NIPR Project Research KP-3, and the Arctic Challenge for Sustainability (ArCS) project. Y. Boose thankfully acknowledges support from the Zeno Karl Schindler foundation. Special thanks to Kim Prather, Andrew Marin and colleagues at the University of California, San Diego for their logistical support during studies at Bodega Bay. We also thank the University of California, Davis Bodega Marine Laboratory for the use of laboratory and office space and shipping and physical plant support while collecting data. We thank the
National Park Service and Prof. Jeffrey Collett for use of their respective mobile laboratories during various field study deployments. We thank Freddie Lamm, Dan Foster, and Marv Farmer at the KSU NW Research Center for their help with coordinating the measurements. Finally, we thank the Nature Conservancy's Canyonlands Research Center and Field Station Manager Philip Adams for helping arrange and select sites for measurements there.

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






Table 1. Samples taken during periods when the CFDC was operated at a single temperature on each date, and immersion freezing methods were aligned in time, sharing samples in some cases. Data from Waverly, CO are from

Garcia et al. (2012). Sample volumes ranged from 1600 to 5500 L in different cases.

| Location | Lat, Lon | Date | Elevation (m) | Standard Sample Type | Instruments |
|---|---|---|---|---|---|
| Waverly, CO | 40.761, -105.076 | 9/29/10 | 1585 | BioSampler | CFDC, IS |
| | | 10/4/10 | | BioSampler | CFDC, IS |
| | | 10/8/10 | | BioSampler | CFDC, IS |
| | | 11/3/10 | | BioSampler | CFDC, IS |
| CSU Atmos Chem, Fort Collins, CO | 40.587, -105.150 | 9/6/13 | 1591 | Ultrapure water | CFDC, IS, CS |
| | | 9/6/13 | | Biosampler blank | CFDC, IS, CS |
| | | 9/6/13 | | Biosampler | CFDC, IS, CS |
| | | 9/6/13 | | 3 μm filter | CFDC, IS, CS |
| | | 9/6/13 | | 0.2 μm filter | CFDC, IS, CS |
| | | 9/12/13 | | Biosampler | CFDC, IS |
| | | 9/12/13 | | 3 μm filter | CFDC, IS |
| | | 9/12/13 | | 0.2 μm filter | CFDC, IS |
| | | 11/12/13 | | Biosampler | CFDC, IS, CS, MOUDI-DFT |
| | | 11/12/13 | | 3 μm filter | CFDC, IS, CS, MOUDI-DFT |
| | | 11/12/13 | | 0.2 μm filter | CFDC, IS, CS, MOUDI-DFT |
| | | 11/13/13 | | Biosampler | CFDC, IS, CS, MOUDI-DFT |
| | | 11/13/13 | | 3 μm filter | CFDC, IS, CS, MOUDI-DFT |
| | | 11/13/13 | | 0.2 μm filter | CFDC, IS, CS, MOUDI-DFT |
| | | 11/14/13 | | Biosampler | CFDC, IS, CS, MOUDI-DFT |






**Table 2.** Sampling locations, elevations, dates and instruments involved in sampling at field sites when the CFDC sampled at varied temperatures during integral offline collections. All sampling at these sites were by filters, except the use of a BioSampler for the CS at Bodega Bay Marine Laboratory and the IS at Waverly, and the MOUDI-DFT at Manitou Experimental Forest (Huffman et al., 2013) and Colby (Mason et al., 2015). CFDC data from Manitou Forest are from Tobo et al. (2013). Data from Waverly, CO are from Garcia et al. (2012).


| Region | Location | Lat, Lon | Date | Elevation (m) | Instruments |
|---|---|---|---|---|---|
| Forest | Manitou Experimental Forest Observatory, CO | 39.094, -105.101 | 8/17/11, 8/18/11, | 2370 | CFDC, MOUDI-DFT |
| Agricultural | Waverly, CO | 40.761, -105.076 | 9/29/10, 10/4/10, 10/8/10, 11/3/10 | 1585 | CFDC, IS |
| Agricultural | Colby, KS | 39.394, -101.066 | 10/14/14, 10/15/14 | 966 | CFDC, IS, MOUDI-DFT |
| Agricultural | Lamont, OK | 36.607, -97.488 | 4/30/14, 5/4/14, 5/5/14, 6/5/14, 6/7/14, 6/8,14 | 315 | CFDC, IS |
| Coastal | Bodega Bay Marine Laboratory, CA | 39.307, -123.066 | 1/26/15, 2/2/15 | 5 | CFDC, IS, CS |
| Semi-arid | Canyonlands, UT | 38.071, -109.563 | 5/11/16, 5/12/16 | 1627 | CFDC, IS, CRAFT |
| Semi-rural | CSU Atmos Chem, Fort Collins, CO | 40.587, -105.150 | 5/18/16, 5/19/16 | 1591 | CFDC, IS, CRAFT |





**Table 3.** Acronyms and symbols (in italics) used

$A_{deposit}$: total area of the sample deposit on the hydrophobic glass cover slip for the MOUDI-DFT method

$A_{DFT}$: area of the sample monitored in the digital video during MOUDI-DFT freezing experiments

AIDA: Karlsruhe Institute of Technology Aerosol Interactions and Dynamics of the Atmosphere cloud chamber

BBY: reference to Bodega Bay, CA, USA field site located at the University of California, Davis Bodega Marine Laboratory

BioSampler: shorthand for impinge device, the Bioaerosol sampler, SKC Inc.

CFDC: Colorado State University Continuous Flow Diffusion Chamber

$C_{INPs}(T)$: number concentration of INPs per volume of liquid

CRAFT: National Institute of Polar Research (Japan) Cryogenic Refrigerator Applied to Freezing Test

CRC: The Nature Conservancy's Canyonlands Research Center, adjacent to Canyonlands National Park, UT, USA

CS: North Carolina State University Cold Stage freezing assay

CSU: Colorado State University, also used to denote the sampling site outside of the Department of Atmospheric Science's Atmospheric Chemistry (Atmos Chem) building

$D_p$: aerosol particle diameter

$f_{ne}$: correction factor to account for the uncertainty associated with the number of nucleation events in each

experiment

$f_{nu}$: correction factor to account for non-uniformity in particle concentration across each MOUDI sample

INP(s): ice nucleating particle(s)

IS: Colorado State University Ice Spectrometer

Kansas: refers to state of Kansas sampling, at Colby, KS, USA

LACIS: Leibniz Institute for Tropospheric Research's Leipzig Aerosol Cloud Interaction Simulator

MEFO: Manitou Experimental Forest Observatory

MOUDI-DFT: University of British Columbia's Micro-Orifice Uniform Deposit Impactor – Droplet Freezing Technique

$n_{INPs}(T)$: number concentration of INPs per volume of air

$n_u$: number of drops unfrozen in immersion freezing arrays

$N$: total number of droplets or liquid entities (in arrays or condensed) in immersion freezing devices

NoCO: Northern Colorado, referring to agricultural sampling region in Waverly, CO

SGP: U.S. Department of Energy, Atmospheric Radiation Measurements program Southern Great Plains site, located near Lamont, OK, USA

STP: standard temperature (273 K) and pressure (1013.5 mb) conditions, typically to refer to volumes converted to these conditions

$T$: Temperature (ºC)

$V$: volume of individual droplets or aliquots in immersion freezing array

$V_s$: sample volume of air collected ($L^{-1}$)

$V_w$: total liquid volume into which particles are placed (mL)



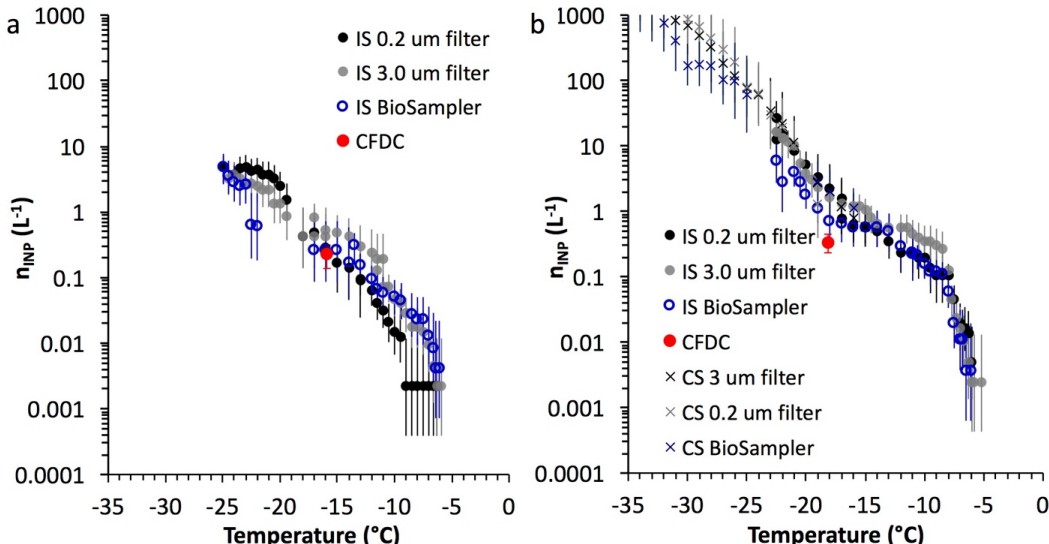

**Figure 1.** Temperature spectra of INP number concentrations ($n_{INP}$) from IS and CS measurements and a CFDC measuring at a single temperature over a 4-hour sampling period. Ambient aerosols were sampled outside of the Colorado State University Atmospheric Chemistry building on a) September 12, 2013; and b) September 6, 2013. Temperature spectra were separately measured for simultaneously collected filter samples with different pore sizes and liquid samples from a BioSampler. Uncertainty values (95% confidence intervals) are shown.





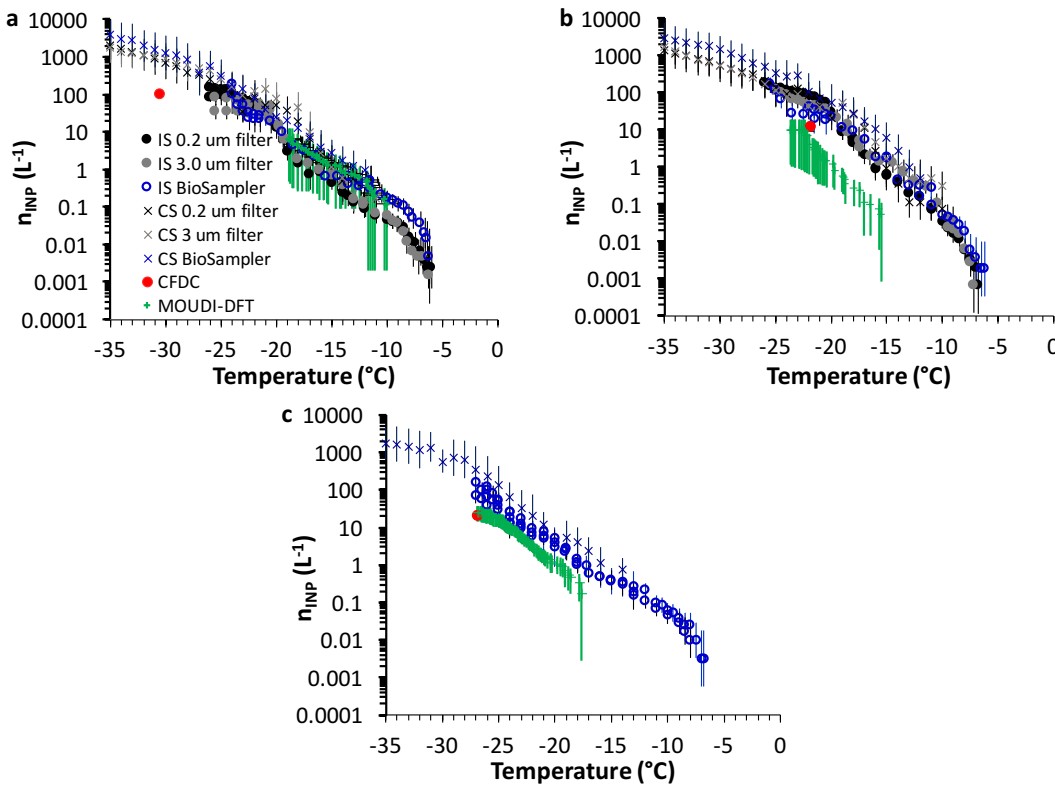

**Figure 2.** Three additional experimental comparison days, as in Fig. 1, but for cases where all four methods were

operational for consistent sampling periods. These dates were November 12 through 14 in panels a), b), c), respectively. The legend is shown in panel a). The additional data in green are from the MOUDI-DFT method (all sizes included), including median (cross), upper and lower bounds.







**Figure 3.** Scatter plot of INP number concentrations obtained with different immersion freezing methods plotted against CFDC online measurement results obtained at 105% RH and temperatures ranging from approximately -15 to -31ºC: (a) IS, (b) MOUDI-DFT (medians of data such as shown in Fig. 1), (c) CS and CRAFT, (d) all data combined from offline immersion freezing tests. The MOUDI-DFT data in (b) include data for all particles sizes assessed ("all") and for the particle size range of 0.3-3.2 μm ("size") best aligned with the effective CFDC sampling size range. Error bars represent 95% confidence intervals, as defined for each method.





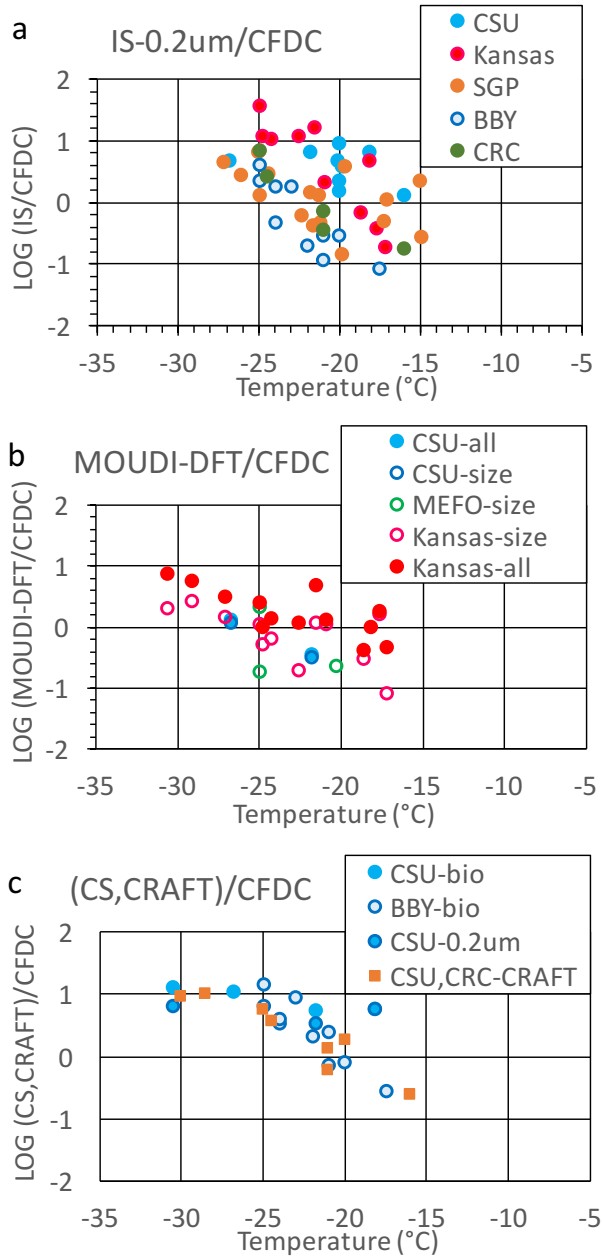

**Figure 4.** Logarithm of the ratio of INP number concentrations measured by various immersion freezing methods versus the CFDC at different sites, denoted as in previous figures. IS 0.2 μm filter samples are shown in a) from five sites. MOUDI-DFT data are compared from three sites in b), where "size" and "all" refer to whether INP number concentrations are from MOUDI size ranges overlapping with sizes permitted into the CFDC or from all sizes. CS and CRAFT ratios are shown in c), where all blue points are for the CS, and "bio" refers to BioSampler collections.





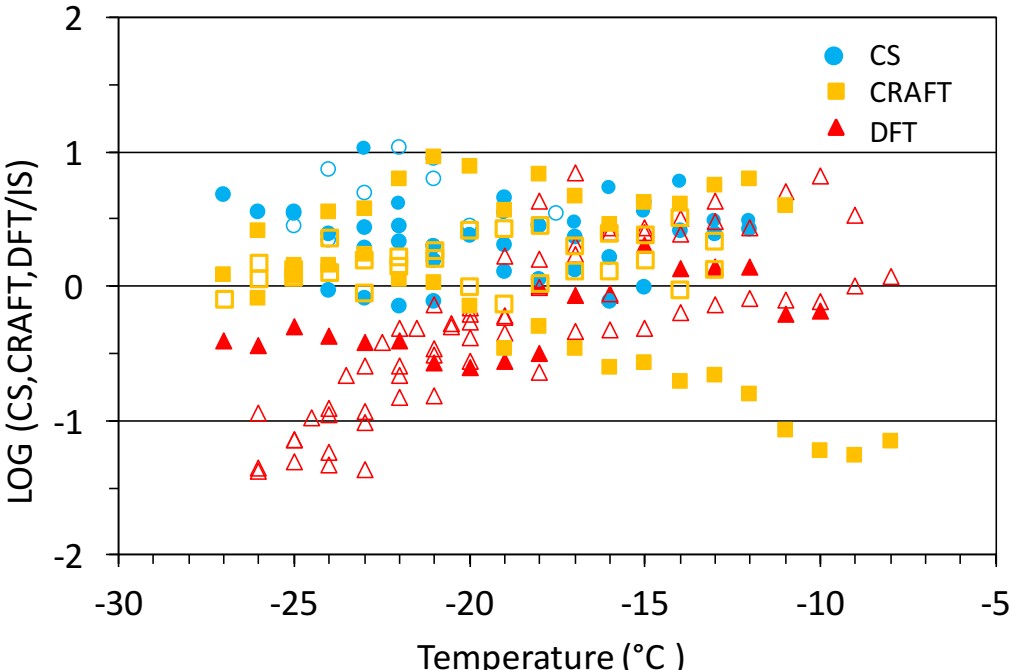

**Figure 5.** Immersion freezing methods comparison, shown as the base-ten logarithm of the ratio of the CS, CRAFT and MOUDI-DFT method INP concentrations for perfect or imperfect overlap of co-sampling periods with the IS INP number concentrations. Samples collected outside the CSU Atmospheric Chemistry facility are shown as filled symbols, while samples collected at other sites on different days (CS: Bodega Bay; CRAFT: Canyonlands Research Center; DFT: Colby, KS) are shown as open symbols.