# Peer review of "Comparative measurements of ambient atmospheric concentrations of ice nucleating particles using multiple immersion freezing methods and a continuous flow diffusion chamber"

_Atmospheric Chemistry and Physics, 2017_

## Referee Comment (RC1) · Anonymous Referee #1 · 30 Jun 2017

In this manuscript, DeMott and co-workers present temperature dependent concentration measurements of atmospheric ice-nucleating particles (INPs) with four offline immersion freezing methods and an online aerosol sampling method. Major goals of the work are to demonstrate the comparability of the different methods at different locations and in different sampling scenarios, and, on this basis, to present and discuss a range of INP measurements. This is a timely and very important topic of interest and impact for the atmospheric sciences community that fits well into the scope of the ACP

journal.

The authors well described the experimental methods and procedures. The manuscript is well structured and written, and it can in principle be accepted and published as is. I only would like to ask the authors to consider the following minor comments for preparing the final version of the manuscript:

Line 362-364: Because in section 2.2 only rough estimates are given for the size-dependent sampling efficiencies of the different techniques, I recommend removing or weaken the statements about INP abundance in different size ranges here.

The black and blue crosses can hardly be distinguished in Figs. 1 and 2. I recommend using other colors or other symbols.

Why are only 1:1 lines shown in Fig. 3? I recommend to also show linear fit lines to the data sets. Why are error bars only shown in panel d of Fig. 3?

Conclusions line 540: I would not say the agreement achieved is excellent. In my view it is good or very good within uncertainty limits.

---

## Referee Comment (RC2) · Anonymous Referee #2 · 4 Jul 2017

In the submitted work DeMott et al. present measurement data for a number of INP collection and immersion freezing methods and compare these to (nearly) contiguous measurements from the CSU CFDC, an instrument that has been running for many years and represents probably the most well understood and calibrated instrument currently in the field. This data is unique in that it attempts to represent sampling done in close temporal and spatial proximity from a number of field locations in the western USA. Furthermore, the work focuses on field data, for which there exists relatively few

representative comparisons for different sampling instruments. Thus the work fits well into the scope of ACP and is an important contribution to the community.

The manuscript is well-written and I believe is publishable "as is" subject to one general comment and a few minor comments that if addressed I believe would add value to the analysis and discussion.

General Comment:

One difficult aspect of digesting the submitted work is that it is at times unclear what instrument specific discussion points can be found in the cited papers (generally dedicated to individual instrument systems) and what is more specific to what is presented in this manuscript. Although, I expect many interested readers have also read the cited literature it is difficult to keep it all at the forefront of ones thoughts. Thus, I would suggest that in revision the authors attempt to more clearly enumerate where instrument specific information can be found in referenced literature and where they are making new statements. For example, the issue of sample storage is raised multiple times but addressed in different ways – it is a challenge to repeatedly return to the literature to see how different instrumental systems have responded to (or not responded to) sample storage and what if any error this introduces.

For some such issues tables, for example including the instrument specific sampling and temperature uncertainties or tolerances, in the text or supplementary material may be beneficial.

Minor Comments:

•lines 215-220 $f_{nu}$ and $f_{ne}$ must be explicitly defined. In the cited literature $f_{nu}$ exists for 2 size ranges and it is unclear what is referred to here. Possibly a combination of the two? Furthermore, at least a sentence or two should be dedicated to an explanation of the origin of these correction factors. This is where the link to the cited material should be provided. Also, please make clear if the parameters used are identical to those

previously published, or are they specific to the particular sample analysis? A short reading of the two papers does not make this evident.

•lines 267-268 See my above example regarding the storage issue. '(not shown)' is a very unsatisfactory parenthetical. Perhaps a better description could be made. e.g., X randomized samples were tested for storage effects by freezing before and after Y days/weeks/months of storage and showed no statistically significant.....

•line 464 'holding for hours at one temperature' The wording is strange here.

•lines 475-490 The discussion of the factor of 3 added as a line in Figure 3 should at a minimum be introduced earlier. Preferably when the figure is introduced. Furthermore, it seems a somewhat deeper discussion of the meaning of this line is missing – that could remain in the discussion. It is clear that the DeMott 2015 et al., paper suggests that this correction factor is used for field measurements of immersion freezing of natural mineral dust for the CFDC – when comparing to a parameterized model of INP. How this relates to the results from other instruments etc. is less clear (e.g., Each of these instruments may have there own c.f. with regard to the DeMott parameterization.). My best understanding is that the 'true' aerosol concentration of (mineral dust) INP as measured by the CFDC should lie somewhere between (inclusive) the 1:1 and factor 3 lines. However, this estimate is also subject to the size limitations of the instrument and parameterization (0.5-2.4 microns). Given the other instruments also operate outside of this range a deeper discussion that ties these links seems warranted. Thus, I also suggest least-squares trendlines be added to the Figure 3 panels or their exclusion defended (For example these trends are essentially explored in Figure 4, but the link is not explicit). Fitting the Figure 3 data by eye, it appears that any trendline would be steeper than the 1:1 line. Is this truly systematic? Are there potentially different explanations for the different instruments? Including at least representative error bars in panels a-c may also assist the discussion.

•Figure 1. Please be explicit (throughout text) with regard to the confidence intervals.

Poisson error, Gaussian?

•Figure S1. Using $n_{inp}$ as the y-axis label maybe confusing. The upper points are actually INP per concentrated liter of sampled air if I understand correctly.

---

## Author Comment (AC1) · 14 Aug 2017

**Response to Anonymous Referee #1:**

We thank Anonymous Referee #1 very much for his/her helpful comments. Below are our point-by-point responses.

**Reviewer's comment [1]:**
Line 362-364: Because in section 2.2 only rough estimates are given for the size dependent sampling efficiencies of the different techniques, I recommend removing or weaken the statements about INP abundance in different size ranges here.

**Authors' response [1]:**
We propose to weaken the existing statements through appropriate wording, but retain a brief discussion that more clearly highlights the basis for the statements. We reference expected capture efficiencies of the filters in Section 2.2, and this should have been reiterated in this results section. These calculated capture efficiencies indicate that the 3-micron pore-size filters should be inefficient at capturing (on the surface or in pores) submicron diameter particles except those well below 0.1 microns in size, while the 0.2-micron pore-size filters have high efficiencies across all diameters. Since the INP concentration results are comparable on the two filter sizes, it suggests a size of INPs in the 1 μm range or larger on average during these sampling periods.

**Changes in manuscript re: comment [1]:**
We rewrite,
*"Considering the capture efficiencies versus size noted in Section 2.2, the lack of significant difference in IS $n_{INPs}$ measured with the filters of 0.2 and 3 μm pore sizes implies that most INPs were likely large enough to be captured effectively. This crudely suggests an INP mode size at about 1 μm or larger. This is also a size that is collected with high efficiency in the Biosampler, for which similar INP concentrations were measured."*

**Reviewer's comment [2]:**
The black and blue crosses can hardly be distinguished in Figs. 1 and 2. I recommend using other colors or other symbols. Why are only 1:1 lines shown in Fig. 3? I recommend to also show linear fit lines to the data sets. Why are error bars only shown in panel d of Fig. 3?

**Authors' response [2]:**
We agree with the reviewer on most of these points. The black and blue crosses have been changed to triangles and a different distinguishing color (gold) is now used for the Biosampler in Figures 1 and 2. We have also added the requested error bars for all panels in Figure 3. The 1:1 line in Fig. 3 is shown as an expectation for perfect agreement. Since we spent two additional figures to discuss the discrepancies between methods as a function of temperature, which in some cases is not linear, we resisted showing linear fit lines in the panels of Fig. 3. The reasoning initially was manifold. First, although these would show a general trend, the fit itself would not add any valuable information on exactly what is going on. The 1:1 line is also the basis for extrapolating perfect agreement on assuming that the CFDC instrument underestimates all natural INPs by the factor that has been reported for mineral dust particles in the laboratory and field. Finally, we spent a great deal of effort in the paper to explain that perfect overlap of samples was a difficult task that requires a lot coordination (and expense on the part of

volunteering groups), with the consequence that only a small amount of data amenable to something like statistical tests was acquitted. When showing data without perfect overlap, the discussion should be a bit more general, as we provide in Fig. 4 and Fig. 5. Nevertheless, since both reviewers have requested these fits, we place them now in addition to the existing lines.

**Changes in manuscript re: comment [2]:**
The new figures appear at the end of this response, as they will be shown in the final article. At the point of introducing these fits in section 3.2 we write:

*"The linear relational slope between IS and CFDC data shown by the light gray dashed line in Fig. 3a. The same representation is applied in all panels of Fig. 3. We provide these fits only to show general trends between the different data sets and do not provide fit parameters herein because a deeper consideration of the source of discrepancies requires additional inspection of trends as a function of temperature, which follows below."*

**Reviewer's comment [3]:**
Conclusions line 540: I would not say the agreement achieved is excellent. In my view it is good or very good within uncertainty limits.

**Authors' response [3]:**
We agree with the reviewer.

**Changes in manuscript re: comment [3]:**
We have modified the sentence accordingly as,
"Very good agreement within uncertainty limits was obtained under…"

[Figure]

Figure 1.

[Figure]

Figure 2.

[Figure]

Figure 3.

---

## Author Comment (AC2) · 14 Aug 2017

**Response to Anonymous Referee #2:**

We thank Anonymous Referee #2 for his/her careful comments that help to improve our presentation. Below are our point-by-point responses.

**Reviewer's general comment [1]:**
One difficult aspect of digesting the submitted work is that it is at times unclear what instrument specific discussion points can be found in the cited papers (generally dedicated to individual instrument systems) and what is more specific to what is presented in this manuscript. Although, I expect many interested readers have also read the cited literature it is difficult to keep it all at the forefront of one's thoughts. Thus, I would suggest that in revision the authors attempt to more clearly enumerate where instrument specific information can be found in referenced literature and where they are making new statements. For example, the issue of sample storage is raised multiple times but addressed in different ways – it is a challenge to repeatedly return to the literature to see how different instrumental systems have responded to (or not responded to) sample storage and what if any error this introduces.

**Authors' general response [1]:**
We appreciate the reviewer's comment and believe that we understand how to fix this, at least in the case noted (storage of samples). However, we are not sure what other specific information is not present that one would need to reference. We acknowledge some difficulty in deciding how to introduce the methods, various comparisons done at different sites, when there was sampling sharing, and so forth. Hence, we decided to introduce the instrumental method first (including how processing of samples occurs and how calculations are made), then special collection considerations, and then the sample sites and objectives in each case. In settling on an approach to organizing this section of the paper, a few things were left a little scattered, and sample storage protocol was one of these. To fix this issue, we have consolidated the discussion of all storage protocol to the same section (Section 2.2). This primarily involved moving the statements about how the UBC group stored their samples. For CSU, NCSU, and NIPR, samples were either collected together and stored frozen always, or frozen in the same manner (excepting -20 versus -80°C at times) by all groups, whether that be a filter or Biosampler sample. It should be understood that similar storage protocol or any issues with storage were not things fully-considered at the start of common sampling that then extended over a few years' time. We were not seeking to recommend protocol but to represent how different groups treat storage.

As discussed by Petters and Wright (2015) in their study of INP measurements from rainwater, the argument that INP activity remains unaltered by the freezing of samples and subsequent storage for some time is at the core of use of immersion freezing methods. They noted the generally better than 1°C repeatability of median population freezing temperatures for droplet suspensions that undergo repeated freeze/thaw cycles [Vali, 2008; Wright et al., 2013, and references therein] in support of their argument. We therefore reference Petters and Wright (2015) and references therein here, and we qualify our statement that our own investigations of this issue indicating (negligible, but not stated here) effects will be covered in a forthcoming paper that can include the type of statistical statement the reviewer would prefer to see. We have added words regarding thawing of samples, and on assumptions that, for the most part, storage

impacts should have been the same, and that this is an additional topic for consideration in future comparisons due to the possibilities of particularly sensitive INPs.

**Changes in manuscript, re: general comment [1]:**
We now write in Section 2.2.:
*"After particle collection, filters were stored frozen at -25 or -80°C in sealed, sterile petri dishes until they could be processed (few hours to few months). Biosampler samples were similarly stored frozen and processed over similar time frames. MOUDI collections for the DFT method were vacuum-sealed after collection, stored at 4°C in a refrigerator, shipping was done with cold packs prior to cold-stage flow cell measurements at the University of British Columbia. We therefore assume similar impacts, if any, of storage on INPs following thawing for processing. This study was not initially conceived as one to test storage impacts on INPs, which should be addressed in future research. We do not expect storage methods to impact result on the basis of existing documentation in the literature. For example, in their study of INPs in rainwater, Petters and Wright (2015) noted that the argument that INP activity remains unaltered by the freezing of samples and subsequent storage for some time is at the core of the general application of immersion freezing methods. They noted, with reference to other literature, the generally better than 1°C repeatability of freezing temperatures for droplets that undergo repeated freeze/thaw cycles."*

In the Conclusions, at the end of paragraph two, we now write:
*"The assumed negligible effect of exact sample storage conditions and the timing of processing after thawing from frozen conditions on INP activity should be inspected more carefully in the future, since some INPs may be susceptible to thermodynamic cycling."*

**Reviewer's general comment [2]:**
For some such issues tables, for example including the instrument specific sampling and temperature uncertainties or tolerances, in the text or supplementary material may be beneficial.

**Authors' general response [2]:**
The reviewer notes some important absent information. We were in fact counting on readers accessing other recent papers that include many of these methods, and statements therein regarding uncertainty and precision that groups attempt to apply uniformly in their work. Our preference now is to insert a synopsis of this information directly into the manuscript under each instrument, and reference to previous publications that include this information. The sections describing each instrument will include some additional details, and we will introduce uncertainty calculations within each section. In all cases, these have remained consistent with what is published.

**Changes in manuscript, re: general comment [2]:**
In the description for the CS we add: *"Temperature uncertainty is based on the manufacturer's (Model TR141-170, Oven Industries) stated tolerance of the cold plate thermistor (± 1°C)."* Regarding confidence intervals, we write, *"Analysis of CS repeat trial data involved binning data into 1°C intervals. Confidence intervals were calculated using two standard deviations of the geometric mean for each bin where multiple data points were available."*

For the IS we add: *"Temperature was measured with 0.1°C resolution and 0.4°C accuracy (Hill et al., 2016)"* For confidence intervals, *"Binomial sampling confidence intervals (95%) were determined for IS data, as described in Hill et al. (2016)."*

The description of MOUDI calculations has been revised as discussed in response to the next comment. We also add, *"Confidence intervals (95%) were calculated based on the Poisson distribution, following Koop et al. (1997). These intervals are nearly equivalent to Binomial confidence intervals for the data in this study."*

For the CRAFT instrument, we add, *"Binomial confidence intervals (95%) were determined, as for the IS data."*

For the CFDC we write, *"CFDC measurement uncertainties vary with processing conditions, and are typically ±0.5°C and 2.4% water relative humidity at -30°C (DeMott et al., 2015)."* Separately, we write,
*"We follow Schill et al. (2016) for correcting sample concentrations for background and for defining confidence intervals for CFDC data, which are represented by error bars in presented plots. Specifically, correct INP concentrations are the sample concentrations with the interpolated background concentrations subtracted. The standard deviation derived from the Poisson counting error during both the sample and the interpolated background concentrations were added in quadrature to obtain the INP concentration error. Concentrations are considered significant if they are 1.64 times larger than the INP concentration error, which corresponds to the Z statistic at 95% confidence for a one-tailed distribution. Consequently, although the lowest limit of detection for 10-min sampling intervals is ~0.2 $L^{-1}$, significant data often requires in excess of 1 $L^{-1}$ INP concentrations."*
We have accordingly revised and tightened up the remaining discussion in Section 2.1 regarding CFDC uncertainties and relation to using the aerosol concentrator.

**Reviewer's minor comment [1]:**
lines 215-220: $f_{nu}$ and $f_{ne}$ must be explicitly defined. In the cited literature $f_{nu}$ exists for 2 size ranges and it is unclear what is referred to here. Possibly a combination of the two? Furthermore, at least a sentence or two should be dedicated to an explanation of the origin of these correction factors. This is where the link to the cited material should be provided. Also, please make clear if the parameters used are identical to those previously published, or are they specific to the particular sample analysis? A short reading of the two papers does not make this evident.

**Authors' response to comment [1]:**
We appreciate the reviewer's pointing out this need to go beyond our simplified explanation.

**Changes in manuscript re: comment [1]:**
To address this comment, we have expanded considerably the description of the correction factors and error analysis associated with the MOUDI-DFT technique. Tables are also added to the supplemental section so that correction factors are explicitly listed. We have revised the previous text listed below:

*"where N is the total number of droplets condensed onto the sample in this case, $A_{deposit}$ is the total area of the sample deposit on the hydrophobic glass cover slip, $A_{DFT}$ is the area of the sample monitored in the digital video during the droplet freezing experiment, V is the volume of air sampled by the MOUDI, $f_{ne}$ is a correction factor to account for the uncertainty associated with the number of nucleation events in each experiment, and $f_{nu}$ is a correction factor to account for non-uniformity in particle concentration across each MOUDI sample (Mason et al., 2015; Mason et al.,2016)."*

This section has been changed to read:

*"where N is the total number of droplets condensed onto the sample in this case, $A_{deposit}$ is the total area of the sample deposit on the hydrophobic glass cover slip, ADFT is the area of the sample monitored in the digital video during the droplet freezing experiment, V is the volume of air sampled by the MOUDI. $f_{ne}$ is a correction factor to account for the statistical uncertainty that results when only a limited number of nucleation events are observed. $f_{ne}$ was calculated following the approach given in Koop et al., (1997) using a 95% confidence interval. $f_{nu}$ is a correction factor to account for non-uniformity in particle concentration across each MOUDI sample (Mason et al., 2015; Mason et al.,2016). This later correction factor consists of two multiplicative terms: $f_{nu,1mm}$ and $f_{nu,0.25-0.10\ mm}$, with these terms correcting for non-uniformity in the particle deposits at the 1mm and 0.25-0.1 mm scale, respectively. Since only a small area (1.2 $mm^2$) of the particle deposits are analyzed and the concentration of particles are not uniform across the entire substrate, $f_{nu,1mm}$ needs to be applied. Since the concentration of particles are not uniform within the small area of the particle deposits analyzed for freezing, $f_{nu,0.25-0.10\ mm}$ needs to be applied. Listed in Tables S3 and S4 are the $f_{nu,1mm}$ and $f_{nu,0.25-0.10mm}$ values applied to the MOUDI-DFT samples collected at CSU and Kansas, respectively. Different correction factors were used for the CSU and Kansas samples since different substrate holders were used to position the glass slides within the MOUDI at the two sites. Substrate holders were not yet employed during the earlier MEFO studies (Huffman et al., 2013). However, using saved slides from the MEFO experiments estimates could be made of the slide offset positions that are needed for defining the non-uniformity correction at the 1 mm scale in Mason et al. (2015). Listed in Table S5 are the $f_{nu,1mm}$ correction factors applied to the MEFO samples based on the slide offset positions. Data were not taken on the non-uniformity within the field-of-view during the freezing experiments ($f_{nu,0.10-0.25\ mm}$) for the MEFO collections, and hence, no correction was applied to the MEFO samples for non-uniformity at the 0.25-0.1 mm scale. On the basis of Mason et al. (2015), cf. Fig. 8 of that paper, and calculations using the factors found for CSU and Kansas sampling, the inability to quantify $f_{nu,0.10-0.25\ mm}$ will lead to an under-prediction of $n_{INPs}(T)$ by a factor that depends on the frozen fraction of droplets at any temperature, perhaps as high as 1.7 for the first drops freezing (1 of ~50-100, or 1-2% frozen fraction) but less than 1.1 once 25% of droplets have frozen."*

**Reviewer's minor comment [2]:**
lines 267-268: See my above example regarding the storage issue. '(not shown)' is a very unsatisfactory parenthetical. Perhaps a better description could be made. e.g., X randomized samples were tested for storage effects by freezing before and after Y days/weeks/months of storage and showed no statistically significant.....

**Authors' response to comment [2]:**
Please see our response to general comment [1]. There is already a strong past basis in the literature for not expecting frozen storage to be an issue for the conditions represented in this paper, and some of the authors are preparing material on this topic for a future paper.

**Reviewer's minor comment [3]:**
line 464: 'holding for hours at one temperature' The wording is strange here.

**Authors' response to comment [3]:**
We agree.

**Changes in manuscript re: comment [3]:**
We have revised the statement to read, "*achieved when droplets remain at a single temperature for periods longer than seconds to minutes*"

**Reviewer's minor comment [4]:**
lines 475-490 The discussion of the factor of 3 added as a line in Figure 3 should at a minimum be introduced earlier. Preferably when the figure is introduced. Furthermore, it seems a somewhat deeper discussion of the meaning of this line is missing – that could remain in the discussion. It is clear that the DeMott 2015 et al., paper suggests that this correction factor is used for field measurements of immersion freezing of natural mineral dust for the CFDC – when comparing to a parameterized model of INP. How this relates to the results from other instruments etc. is less clear (e.g., Each of these instruments may have their own c.f. with regard to the DeMott parameterization.). My best understanding is that the 'true' aerosol concentration of (mineral dust) INP as measured by the CFDC should lie somewhere between (inclusive) the 1:1 and factor 3 lines. However, this estimate is also subject to the size limitations of the instrument and parameterization (0.5-2.4 microns). Given the other instruments also operate outside of this range a deeper discussion that ties these links seems warranted. Thus, I also suggest least-squares trendlines be added to the Figure 3 panels or their exclusion defended (For example these trends are essentially explored in Figure 4, but the link is not explicit). Fitting the Figure 3 data by eye, it appears that any trendline would be steeper than the 1:1 line. Is this truly systematic? Are there potentially different explanations for the different instruments? Including at least representative error bars in panels a-c may also assist the discussion.

**Authors' response to comment [4]:**
We disagree on this point. The *cf* clearly relates only to the CFDC and it is used in figures where other method results are compared to the CFDC. No one has explored if there are biases involved in measuring maximum INP number concentrations via other methods, but we do provide discussion of such possibilities in this paper. Bulk suspension immersion freezing methods are intended to capture the full aerosol size distribution and not to "miss" INPs, but other factors may come into play when a population of aerosol particles are placed into liquid. Relating different measurement methods is a multivariate problem involving limitations and potential artifact influences in all instruments, a problem for which we have tried to offer insights and a path forward for full investigation.

The reviewer's understanding of the DeMott et al. (2015) parameterization is not correct. The fit to determine *cf* for mineral dust particles exclusively (excluding arable soil dusts, for example) used INP activation data at higher RH to determine cf versus more typical RH processing values that need to be used in sampling atmospheric INPs in a regime where particles activate as water droplets prior to freezing. The value varies around cf = 3, not between cf =1 and cf = 3. This is the value that is justified for the CSU CFDC. We reference another paper on this topic and its potential source that notes 2-10 factors. Therefore, this appears a common issue with similar instruments. Until more is known, including the appropriateness of these factors for ambient INPs (likely), we wish to retain mention of the factor within the discussion section of the paper. Finally, the parameterization is applied to all sizes above 0.5 microns, not just the ones the CFDC samples. This is a common misconception, but the publication is quite clear on the use of all particles above 0.5 μm, regardless of whether the CFDC can capture all of these or not. The reason for that was practical, for ease of application in modeling studies. Consideration for missing INPs at larger sizes was left for future study. We feel that we have otherwise provided sufficient discussion of potential size limitation effects, at least as can be warranted in this first inspection of sampling ambient INPs by different techniques. Advice is offered for future focused comparisons.

While we do not feel that a regression fit to the data in Fig. 3 adds to the demonstration of discrepancies provided in Fig. 4 and Fig. 5, we include these lines now. Are the trends systematic? Not across methods. Is that meaningful? We are not sure yet, although we have offered a number of reasons why it might be, at least for the type of sampling conducted in this paper. Are there other explanations we have not thought of? That seems a topic for future research. The regression fits in each panel of Fig. 3 now show a general trend for each method versus the CFDC, although we point out that the imposition of this fit adds no special valuable information regarding exactly what is going on. In contrast, the 1:1 line in Fig. 3 is meaningful as the basis for extrapolating perfect agreement of the CFDC with other measurements, and the 3*CFDC line is meaningful for exploring whether the CFDC instrument underestimates all natural INPs by the factor that has been reported for mineral dust particles in the laboratory and field. Figures 4 and 5 are used to discuss the discrepancies between methods as a function of temperature, and these inspections reveal discrepancies that are not truly linear or understood from the regressions in Fig. 3. Finally, we spent a great deal of effort in the paper to explain that perfect overlap of samples was a difficult task that requires a lot coordination on the part of groups volunteering their effort in this case, with the consequence that only a small amount of data truly appropriate for statistical tests was acquired. When showing data without perfect temporal overlap, we feel that the discussion should be a bit more general. Figures 4 and 5 provide this. This paper is the start toward what would be necessary for more exact comparisons in the future.

**Changes in manuscript re: comment [4]:**
The new figures appear at the end of this response, as they will be shown in the final article. At the point of introducing these fits (first mention of trend lines in section 3.2) we write:

*"The linear relational slope between IS and CFDC data shown by the light gray dashed line in Fig. 3a. The same representation is applied in all panels of Fig. 3. We provide these fits only to*

*show general trends between the different data sets and do not provide fit parameters herein because a deeper consideration of the source of discrepancies requires additional inspection of trends as a function of temperature, which follows below."*

**Reviewer's minor comment [5]:**
Figure 1. Please be explicit (throughout text) with regard to the confidence intervals. Poisson error, Gaussian?

**Authors' response to comment [5]:**
Done.

**Changes in manuscript re: comment [5]:**
More detail describing the confidence intervals is provided now in the discussion of the individual instruments, and consolidated in Section 2.

**Reviewer's minor comment [6]:**
Figure S1. Using $n_{inp}$ as the y-axis label maybe confusing. The upper points are actually INP per concentrated liter of sampled air if I understand correctly.

**Authors' response to comment [6]:**
We thought that the description in the caption might be enough to make this clear to readers, but have amended that as best possible. The axis should be consistent, in that it is always a number per unit volume, concentrated in one case and not in the other.

**Changes in manuscript re: comment [6]:**
We rephrase the caption as, "*Aerosol Concentrator calibration check at -30℃ at CSU on May 19, 2016. This is a typical experimental sampling period at one temperature. In this figure, the lower data points are INP number concentrations without using the aerosol concentrator. Alternating periods of high INP number concentrations are during use of the aerosol concentrator. Inspection of the ratio between the INP number concentrations per volume of air during periods on versus off the concentrator reveal the CF factor, which is ~90 in this case. The shaded lower region is the limit of significance for INP concentrations, as described in Section 2.1 of the manuscript.*"

[Figure]

Figure 1.

[Figure]

Figure 2.

[Figure]

Figure 3.